# Unlocking Translational Potential: Conditionally Reprogrammed Cells in Advancing Breast Cancer Research

**DOI:** 10.3390/cells12192388

**Published:** 2023-09-30

**Authors:** Danyal Daneshdoust, Mingjue Luo, Zaibo Li, Xiaokui Mo, Sahar Alothman, Bhaskar Kallakury, Richard Schlegel, Junran Zhang, Deliang Guo, Priscilla A. Furth, Xuefeng Liu, Jenny Li

**Affiliations:** 1Comprehensive Cancer Center, Ohio State University, Columbus, OH 43210, USA; 2Departments of Pathology, Wexner Medical Center, Ohio State University, Columbus, OH 43210, USA; 3Department of Biostatics and Bioinformatics, Wexner Medical Center, Ohio State University, Columbus, OH 43210, USA; 4Departments of Oncology and Medicine, Lombardi Comprehensive Cancer Center, Georgetown University, Washington, DC 20057, USA; 5Departments of Pathology, Lombardi Comprehensive Cancer Center, Center for Cell Reprogramming, Georgetown University, Washington, DC 20057, USA; 6Department of Radiation Oncology, Wexner Medical Center, Ohio State University, Columbus, OH 43210, USA; 7Departments of Pathology, Urology, and Radiation Oncology, Wexner Medical Center, Ohio State University, Columbus, OH 43210, USA

**Keywords:** conditionally reprogrammed cells, breast cancer, precision medicine

## Abstract

Preclinical in vitro models play an important role in studying cancer cell biology and facilitating translational research, especially in the identification of drug targets and drug discovery studies. This is particularly relevant in breast cancer, where the global burden of disease is quite high based on prevalence and a relatively high rate of lethality. Predictive tools to select patients who will be responsive to invasive or morbid therapies (radiotherapy, chemotherapy, immunotherapy, and/or surgery) are relatively lacking. To be clinically relevant, a model must accurately replicate the biology and cellular heterogeneity of the primary tumor. Addressing these requirements and overcoming the limitations of most existing cancer cell lines, which are typically derived from a single clone, we have recently developed conditional reprogramming (CR) technology. The CR technology refers to a co-culture system of primary human normal or tumor cells with irradiated murine fibroblasts in the presence of a Rho-associated kinase inhibitor to allow the primary cells to acquire stem cell properties and the ability to proliferate indefinitely in vitro without any exogenous gene or viral transfection. This innovative approach fulfills many of these needs and offers an alternative that surpasses the deficiencies associated with traditional cancer cell lines. These CR cells (CRCs) can be reprogrammed to maintain a highly proliferative state and reproduce the genomic and histological characteristics of the parental tissue. Therefore, CR technology may be a clinically relevant model to test and predict drug sensitivity, conduct gene profile analysis and xenograft research, and undertake personalized medicine. This review discusses studies that have applied CR technology to conduct breast cancer research.

## 1. Breast Cancer and Clinical Challenges

Breast cancer is the second most common type of cancer among women in the world, with an estimated 2.3 million new cases accounting for 11.7% of all cancer cases and approximately 700,000 deaths worldwide [1,2]. In the United States alone, more than 300,500 new cases and 43,700 deaths are reported annually [3]. Breast cancer is classified into four major molecular subtypes, namely luminal A (Estrogen Receptor (ER) and/or progesterone receptor (PR)-positive, HER2-negative, Ki-67<14%), luminal B ([luminal B (HER2-negative): ER and/or PR-positive, HER2-negative, Ki-67 >14%] [luminal B (HER2-positive): ER and/or PR-positive, HER2-over expressed or amplified, any Ki-67), HER2-enriched (ER/PR-negative, HER2-positive), and basal/triple negative (ER/PR/HER2-negative) based on human epidermal growth factor receptor-2 (HER2) receptor and hormonal status [4,5]. The management of breast cancer patients relies on the assessment of hormone receptor status, specifically PR, ER, and HER2 [6,7]. Hormonal therapies have proven effective for the majority of patients with hormone receptor-positive subtypes. For instance, selective estrogen modulators (SERMs) such as tamoxifen or selective estrogen receptor degraders (SERDs) such as fulvestrant, or aromatase inhibitors, sometimes in conjunction with CDK4/6 inhibitors, are used for ER-positive breast cancer, while the combination of Trastuzumab (Herceptin), Docetaxel (Taxotere), and Pertuzumab (Perjeta) can be employed for HER2-positive breast cancer [8,9,10,11]. Nevertheless, addressing the issue of both primary and acquired resistance to hormonal therapies continues to be a challenge [12]. Given the significant burden of cancer, researchers worldwide continue to strive toward understanding tumor growth and treatment outcomes in breast cancer using cutting-edge approaches. 

A major obstacle hindering cancer research progress is the limited availability of cancer models [13]. Cancer cell lines have been widely used as effective models for drug discovery and preclinical studies. However, depending on the type and stage of the disease, the success rate for producing cancer cell lines is as low as 1–10% [14]. Although the number of available cell models has been increasing, they are still not enough to effectively study various types of cancers. Moreover, traditional cancer cell lines have a limitation in replicating the intricate heterogeneity of primary tumors, which considerably restricts the advancement of basic and translational medicine [15]. Animal models are widely used in laboratory cancer research and have made significant contributions to our understanding of the biology of cancers [16]. Additionally, they have played a crucial role in preclinical studies of different types of cancers. In breast cancer, these models can take various forms, such as based on chemical carcinogenesis, genetically modified animals, xenograft models, syngeneic models, and other approaches and tools [17,18,19]. Animal models are frequently utilized in cancer research as a stand-in for humans. However, due to variations in genetics and biology between different species, the translation of experimental findings from animal models to clinical practice can be slow [20]. Creating feasible and innovative models for translational medicine and cancer research is crucial to tackling these challenges. 

Recent progress in biotechnology has led to significant changes in the development of cancer models. Patient-derived models (PDMs) have emerged as a particularly promising approach, as they maintain consistent genetic characteristics with their parental tumors. Different types of PDMs such as organoids, induced pluripotent stem cells (iPSCs), patient-derived xenografts (PDXs), and conditionally reprogrammed cells (CRCs) have been widely used in cancer research due to their ability to better replicate the complexity of human tumors [20,21,22,23,24,25]. PDMs have different uses in cancer research depending on the context and methods employed. A comparison of these models (Table 1) is presented in this review, along with a detailed discussion of the conditional reprogramming (CR) technology. This article provides a comprehensive review of the current state and potential applications of the CR method in primary mammary epithelial cells and breast cancer research.

## 2. Patient-Derived Cancer Models 

### 2.1. Organoids

Organoids are three-dimensional (3D) cell structures derived from primary tissues or stem cells that mimic the architecture and function of the original organ [26]. The concept of 3D culture emerged in the 1980s [27,28]. During the exploration of 3D culture techniques, Emerman et al. conducted pioneering experiments utilizing normal mammary epithelial cells and collagen gels. Their findings highlighted the remarkable advantages offered by floating collagen gel substrates within a three-dimensional environment. Notably, they demonstrated that these substrates provided unique growth and structural differentiation factors for mammary epithelial cells, surpassing the capabilities of conventional plastic substrates [28]. Similarly, various studies conducted during the 1980s reaffirmed the concept that mammary myoepithelial cells can organize themselves when cultured in collagen gels [29,30,31]. In the year 1992, Petersen et al. introduced the term “organoids” to define the well-organized structures that emerge from 3D cultures [32]. In 2007, the Bissell group outlined two protocols for cultivating normal and cancerous human mammary cells in the 3D structure [33]. The cells were cultured by either allowing them to grow or embedding them in Matrigel, with durations of 10 days or 4 days, respectively. Sato et al. [34] developed long-term cultures of intestinal organoids. After being initially established for the intestine, their technique was later adapted for use in other organs such as the prostate, colon, lung, stomach, pancreas, liver, and most recently the breast. This resulted in the creation of a biobank of organoids, which can be used for drug screening and molecular research [34,35,36,37,38,39,40]. Breast organoids have been utilized to investigate factors that impact signaling transduction, tissue remodeling, and gene expression [41]. Using mouse mammary epithelial organoids derived from both a normal mouse mammary epithelial cell line and 10-week-old CD-1 mouse tissue, Simian et al. proposed the significant involvement of matrix metalloproteinases in mammary branching morphogenesis [42]. Sumbal et al. demonstrated that the branching of mammary epithelium is regulated by fibroblasts using epithelial organoids isolated from the mammary glands of pubertal mice [43]. In a similar vein, Zhang et al. demonstrated the involvement of different FGF ligands in regulating epithelial behavior using mouse mammary organoids [44]. During their investigation of FGF receptors in an immortalized murine mammary epithelial cell line, Xian et al. uncovered the potential of 3D culture in comprehending the tissue’s response to growth factors [45]. Additionally, the application of 3D culture has proven to be valuable in investigating different types of progenitor cells within the breast. This has been revealed by studies employing normal human mammary epithelial organoids [46]. Davaadelger et al. examined breast organoids derived from BRCA1 mutant human mammary tissue. These organoids were subsequently treated with progesterone and estradiol, revealing distinct differences in progesterone receptor activity compared to non-carrier organoids, confirming results initially reported in genetically engineered mice with loss of Brca1 over a decade earlier [47,48]. Sachs et al. successfully established over 100 primary and metastatic breast cancer organoid lines derived from human breast cancer tissue. Remarkably, the majority of these breast cancer organoids closely resembled their corresponding original breast cancer tumor in terms of histopathology, HER2 receptor status, and hormone receptor status [36]. In a recent study, Dekkers et al. employed breast organoids derived from normal breast tissue and genetically modified them using CRISPR/Cas9 technology to create a model of breast cancer. By knocking out four BC-associated tumor suppressor genes using CRISPR/Cas9, they successfully recapitulated the process of oncogenesis. They demonstrated long-term culturing ability and responsiveness to therapeutic interventions [49]. 

Despite the organoid system’s promise, it has some drawbacks and technological challenges. It is significant to note that numerous phases of differential centrifugation and cell straining are used in many contemporary techniques to separate and purify mammary epithelial cells from surrounding stromal cells before growth in 3D culture. Breast organoids made from pure stromal and epithelial cells might not accurately replicate the in vivo tissue architecture and cell-cell/matrix interactions [50]. Although only epithelium is required in the widely recognized definition of organoids, the inclusion of additional cell types may help to more accurately recreate in vivo breast architecture.

### 2.2. Patient-Derived Xenografts

PDX models are generated by transplanting fragments of human tumors into mice with immunocompromised immune systems [51,52]. Over the past few years, PDXs have successfully recapitulated the genetic characteristics, gene expression patterns, and tissue histology of the original tumor [53]. They are now widely acknowledged as a preclinical model that closely mimics physiological conditions. In breast cancer studies, PDX models rely mainly on implanting small 1 mm3 fragments from patient tumors into the subcutaneous tissue or mammary fat of immunocompromised mice. The tumors that develop can be further propagated in mice, and even after multiple passages in the mouse model, the grafts maintain their characteristic phenotype and genetic stability. They continue to exhibit essential histological and molecular characteristics of the original tumors, including their potential to spread and form metastases [54,55,56]. However, it has been recently observed that copy number alterations can emerge during the process of PDX passaging. This may be attributed to the selection of minor clones that were not present in the original patient tumors during their evolution [57]. Furthermore, PDX models have demonstrated treatment responses that closely resemble those observed clinically [58]. Therefore, they serve as a valuable tool for investigating tumor heterogeneity, metastasis, and conducting preclinical drug testing. Additionally, they enable the examination of signaling pathway activation before and after the occurrence of drug resistance.

While PDX tumor models have many advantages, they also have some drawbacks, such as the possibility for selection of minor clones as mentioned above. Moreover, creating PDX models may be an expensive and time-consuming process that takes anything from six months to two years. In addition, depending on the specific characteristics of the disease and the origins of the tumor, the success rate of developing PDX models might vary greatly, ranging from 10% to 90% [52,59]. The quick loss of human stromal components in PDXs, which are replaced by the murine microenvironment upon engraftment, is another drawback [60]. Changes in the tumor’s paracrine regulation and its physical characteristics, such as interstitial pressure, may result from the transition to the murine stroma, which may limit research on anticancer agents that target this particular tumor compartment [61]. Finally, PDX models rely on immunodeficient hosts, leading to a lack of essential immune cells and an inability to fully replicate the response of the human immune system to tumors and tested drugs. To address this limitation, humanized mice with reconstituted human immune systems have been developed to provide a unique platform for studying human immune responses and evaluating immune-based therapies [52,62]. Hence, the aforementioned limitations have impeded PDX models from offering practical references for clinical decision-making. There are still important challenges to be addressed to make this platform more informative.

### 2.3. Conditionally Reprogrammed Cells

Identifying a single model system that is fast, easy to perform, and has a high success rate from a variety of clinical samples (surgical samples, needle biopsy, cryopreserved tissue, blood, urine) remains a challenge. We have recently developed a new primary cell culture system called conditional reprogramming. To efficiently and rapidly produce infinite cells, we added irradiated Swiss-3T3-J2 mouse fibroblast cells and Y27632, a Rho-associated kinase (ROCK) inhibitor, to the cell culture plate using this method [63]. Conditionally reprogrammed cells (CRCs) is the term used to describe cells produced using this approach. From a variety of tissues, including those obtained through fine-needle aspiration, core biopsies, surgical specimens, and patient-derived xenograft tissues, the CR technique may effectively produce huge numbers of primary cells [64]. The genetic and histological properties of the parent tissue can be recapitulated in CRCs, allowing them to maintain a highly proliferative state [65]. It is interesting to note that the phenotype is completely reversible if these factors are eliminated [66]. Therefore, CR technology can be an ideal model for performing gene profiling analysis [65,67] and drug sensitivity testing for use in breast and other cancer research [68], regenerative medicine, and xenograft studies. An important feature is that these cultures can be used for establishing xenografts [69], patient-derived xenograft cell lines [70], cell cultures from PDXs, and organoid cultures. Finally, conditionally reprogrammed cells maintain cell lineage commitment and the cell heterogeneity found in a biopsy [63,69,71,72,73,74].

The ease of execution, genotype stability, exponential growth, and high success rate in a single model system are some of the reported advantages of CR. These properties qualified CR as an outstanding in vitro model compared to other models such as organoids and patient-derived xenografts [64]. Clinical research applications of CR technology have been investigated in breast cancer [75], lung cancer [76], prostate cancer [71], bladder cancer [77], gastric cancer [78], liver cancer [79], and salivary cancer (68). 

The conditional reprogramming system can be successfully applied to various epithelial tissues such as prostate, kidney, skin, lung, and several others [65,80]. Moreover, the use of the system is not limited to humans. It can be applied to many kinds of mammals such as mice, rats, horses, dogs, and cows [81]. Recent publications summarized applications of CR in cell biology, virology, and cancer research [24,64,66,77,78,82,83,84,85]. In this article, we review studies that applied CR technology for breast cancer research (Figure 1).

**Table 1 cells-12-02388-t001:** Comparisons between patient-derived models: 2D culture, organoids, patient-derived xenografts (PDXs), and conditionally reprogrammed cells (CRCs).

Methods	Advantages	Shortcomings	References
2D culture	1. Inexpensive technique 2. Easy to manipulate genetically 3. Suitable for high-throughput drug screens in a short amount of time at a low cost	1. Loss of tumor heterogeneity 2. Genetic drift between different laboratories (for cell lines) 3. Lack of microenvironment 4. Not suitable for low-grade tumor establishment 5. limitation of cell-cell and cell-extracellular matrix interactions	[14,15]
Organoids	1. 3D culturing 2. Can generate both healthy and tumor organoids 3. Maintain tumor heterogeneity 4. Possibility to co-culture tumor organoids with elements of the microenvironment (pathogens [bacteria] and immune cells)	1. Lack of microenvironment (immune cells, vasculature, and microbiota) 2. Dependent on stem cells 3. Lack of protocol and medium standardization 4. Overgrowth of nonmalignant cells	[34,35,36,37,38,39,40]
PDXs	1. In vivo model 2. Direct engraftment from human tumor 3. Maintain histological, genomic, and transcriptomic features of tissue of origin 4. Recapitulate the natural environment of the tumor 5. humanized mice model with reconstituted human immune systems	1. Expensive technique 2. Resource and time consuming 3. Not suitable for high-throughput drug screening 4. Rely on interactions with a mouseMicroenvironment 5. Only tumor models 6. challenging to be reproducible on a large scale	[52,55,58,61]
CRCs	1. A wide range of specimen sources 2. Paired normal and tumor cells culturing 3. Cost saving and rapid expansion (1–10 days) 4. Can maintain original karyotype and tumor heterogeneity 5. High-throughput drug screening6. Gene profiling analyses 7. Suitable for low-grade tumor establishment	1. Contamination with feeder cells 2. Overgrowth of benign cells 3. Lack of stromal components	[63,64,65,66,67,68]

## 3. Applications of CR in Primary Mammary Epithelial Cells

The adult human mammary gland consists of a complex structure made up of tubuloalveolar units. These tubuloalveolar units are composed of polarized epithelial cells responsible for producing milk, which is surrounded by myoepithelial cells. The overall organization of the gland is characterized by a two-layered arrangement, with a basement membrane encompassing the entire structure [86]. Understanding the function of the mammary gland in animal models has substantially advanced our understanding of gene regulation [87], hormone action [88], and stem cell biology [89] during the past few decades. However, using knowledge gleaned from animal models to explain the evolution of human mammary glands is not optimal. There are many differences between the mammary glands of rodents and humans, including their anatomical locations, histological compositions, and gene expression profiles [90,91]. Furthermore, it is challenging to separate and examine molecular events at the cellular level as a result of whole-animal research [92]. The mammary epithelium biology has been extensively studied in vitro using cultured immortalized human breast cell lines such as MCF-10A. However, it has lately been questioned whether these cells are suitable in vitro models for human mammary epithelial cells [93]. According to studies, continuous cell lines demonstrate increased lineage-restricted profiles that make them unable to accurately recapitulate the intratumoral heterogeneity of different breast tissues [94]. For instance, EpCAM+CD24+CD49f+ and EpCAM+CD49f- populations are lost in normal breast cell lines compared to primary breast epithelial cells obtained from reduction mammoplasty. Additionally, mammary cell lines such as HME I/II and MCF-10A are unable to develop into mature luminal breast epithelial cells despite maintaining the characteristics of bipotent progenitor cells [94]. Hence, it is desirable to develop in vitro models that more accurately mimic the physiologically relevant heterogeneity of the epithelial cells in the tissue of the human mammary gland.

Using primary epithelial cells directly derived from human mammary glands offers a tissue-specific model; however, it has certain limitations. One of the limitations is that these cells have a short lifespan when cultured under conventional tissue culture conditions [95]. CR system demonstrated that feeder cell-conditioned medium, together with a Rho-associated kinase (ROCK) inhibitor, can induce inexhaustible and rapid in vitro proliferation of primary epithelial cells from normal and malignant breast tissues [63,65,67,69,80,96]. 

Jin and colleagues investigated the feasibility of the CR system for the characterization of primary human mammary epithelial cells (PHMEC) isolated and propagated by conditionally reprogrammed cell culture [97]. They established cell cultures of PHMEC based on the Liu et al. procedure [80]. They have shown that these cells maintain numerous important characteristics associated with mammary tissue cells. The cultured cells exhibit the expression of markers typically found in luminal and myoepithelial cells. Interestingly, some conditionally reprogrammed cells express both luminal and myoepithelial markers, making their cell identity unclear. The persistent presence of CK19 staining in the cultured cells suggests that this model can retain specific characteristics inherited from primary tissues, particularly in the early passages. CD49f and EpCAM are commonly utilized to categorize subpopulations of mammary epithelial cells. Myoepithelial cells and potential bipotent progenitor cells are identified by a CD49f+EpCAM-/low phenotype, whereas Luminal progenitor cells are typically characterized by a CD49f+EpCAMhigh profile. They found four distinct subpopulations, including CD49f+EpCAMhigh and CD49f+EpCAM-/low cells, that were each characterized by a different amount of EpCAM and CD49f expression. Additionally, cells with the CD49f- EpCAMhigh profile, a marker for mature luminal cells, were seen. Therefore, they considered that the conditional reprogramming strategy will preserve the mammary epithelial cells’ complex heterogeneity. They demonstrated that the CR technique enables the in vitro establishment of heterogeneous cultures from normal human breast tissue. Importantly, these cultures showed Erα expression because estrogen stimulates Erα function. Both the construction of different 3D organoid structures and the development of milk-producing cells are supported by this culture technique. Alothman et al. established PHMEC to study the behavior and transcriptomes of non-cancerous human mammary epithelial cells at risk for breast cancer development [67] (Figure 2). 

They showed that PHMEC initially isolated using CRC could be transferred and grown in a variety of different mammary cell specific media. Then, exploiting the ability of utilizing CRC-isolated cells for parallel studies of behavior and gene expression, showed that estrogen growth response was associated with tissue necrosis factor signaling and interferon alpha response gene enrichment while neoadjuvant chemotherapy exposure significantly altered transcriptomes, shifting them towards expression of genes linked to mammary stem cell formation. Finally, they showed gene expression patterns in mammary cells normally linked to pregnancy can be found as an abnormal finding in non-pregnant at-risk and breast cancer cells [67]. Alamri et al. used genetically engineered mice carrying Brca1 and Trp53 mutations to explore how CRC methodology compared to his- torical methods of mammary epithelial cell isolation for initial primary cell isolation, allograft generation, and impact of CRC and passage on the transcriptome [65]. They compared the CR system with mammary-optimized EpiCult™-B (EpiC) for isolating and propagating primary mammary epithelial cells, generating allografts, and examining the genome-wide transcriptional effects. They conducted their investigation using both cancerous and non-cancerous mammary tissue from mice with varying levels of BRCA1 and p53. They showed the high success rate of CRC in the initial isolation and propagation of primary cells. Mammary epithelial cells, initially isolated using CRC, subsequently could be transitioned to a different culture method. One notable advantage of CRC is that the growth of epithelial cells is selectively enhanced under the conditions used in their study. As a result, implementing CRC for the initial isolation process not only increased the success rate of cell cultures but also reduced fibroblast contamination. They concluded that CRC is more efficient for the initial generation of cultures, while EpiC is more suitable for allograft generation [65]. Saenz and colleagues conducted a study to examine the feasibility of generating mouse mammary epithelial CRCs using either normal or tumor tissues [98]. Furthermore, they sought to determine whether the characteristics of these mouse mammary epithelial CRCs resembled those of human cells when exposed to the CRC system. While mouse epithelial cells exhibit senescence over several passages, the mechanisms underlying senescence differ from those observed in human cells. Notably, telomere shortening does not appear to be a significant contributor to the senescence process in mouse cells. Interestingly, despite these differences, Saenz et al. revealed that both normal and tumor-derived mammary epithelial CRCs from mice could be continually passaged without limitations. They observed that similar to human epithelial cells, normal mouse mammary epithelial CRCs displayed the presence of markers associated with progenitor cells while lacking pluripotent stem cell markers. Furthermore, when cultured in a 3D Matrigel matrix, mammary epithelial CRCs were capable of forming mammary acinar structures. In contrast to human cells, the study revealed that mouse mammary epithelial CRCs maintained high levels of expression for numerous progenitor cell markers even after the withdrawal of the CRC system. This suggests that in mouse cells, the effects induced by the CRC system are not rapidly reversible. This is one of the notable differences between mouse and human CRCs. Mammary epithelial CRCs derived from mouse mammary tumors obtained from MMTV-Neu mice were also found to be capable of indefinite passage. Furthermore, a substantial proportion of these cells exhibited the expression of markers typically associated with tumor-initiating cells when examined in vitro. When MMTV-Neu mammary epithelial CRCs were transplanted into mice, they were capable of forming tumors that closely resembled the tumors and metastases observed in MMTV-Neu transgenic animals. Histopathologically, these tumors were indistinguishable. They noted a substantial overlap in the properties of mouse and human CRCs based on these findings, suggesting similarities between mouse and human CRCs. Therefore, the CRC technology can be effectively employed with both normal and transformed mouse epithelial cells, providing valuable opportunities to investigate the properties of genetically manipulated cells in allograft models [98]. This highlights the potential of the CRC system as an ideal tool for studying various aspects of mouse epithelial cell behavior and characteristics and that the CRC method may offer an ideal tool for studying the function of mammary cells and the factors that influence malignant transformation [97].

## 4. Applications of CR in Breast Cancer Research

In addition to applications on basic mammary epithelial biology, recent studies suggested CR technology also can be used for modeling human breast cancer (Figure 2), biology of cancer disparity, molecular heterogeneity of breast cancer, and clinical translations (precision diagnostics and treatment) (Table 2). 

### 4.1. Modeling Diseases 

Efficient and rapid establishment of CRCs has been achieved using various human normal and tumor specimens, including the breast, without the need for exogenous viruses or genetic manipulation. These CRCs can preserve the characteristics of their primary tissues. By removing these conditions, the cells’ differentiation ability can be restored. They also can be cultured in both 2D and 3D systems. Consequently, CR can serve as an ideal in vitro model for studying breast cancer. Mahajan et al. applied CR technology to conduct a comparative analysis between early-passage conditionally reprogrammed breast cancer cells and their corresponding primary tumors [75]. They evaluated the genomic characteristics of six newly established CRC cultures derived from invasive breast cancer and compared them to the original primary breast tumors. Simultaneous profiling of CRCs and their corresponding primary breast tumors was conducted using targeted next-generation sequencing, genome-wide array-CGH, and global miRNA expression analysis. This comprehensive approach aimed to determine the molecular similarities between the two in terms of gene mutations, copy number alterations (CNAs), and miRNA expression levels. Using flow cytometry, they also evaluated the ploidy and amount of epithelial cells in the CRCs. The study findings indicated that the CRCs retained the overall genomic signatures of the original primary breast tumor and demonstrated a similar pattern and level of CNAs. The array-CGH analysis showed a high level of overlap, ranging from 72 to 100%, between the CRCs and their corresponding primary breast tumors. Notably, the cytobands commonly affected by CNAs displayed more than 95% overlap between each CRC and its corresponding primary breast tumor. Furthermore, the copy number profiles of these CRCs exhibited non-random and recurrent CNAs that are typically associated with specific intrinsic subtypes of breast cancer. The analysis using targeted next-generation sequencing also showed that the established CRCs maintained the particular gene alterations that had been found in their original tumors. Analysis of three paired CRCs and primary breast tumors (Cases 2, 4, and 6) revealed that they share the same type of variants affecting the FLT3, TP53, CDKN2A, PIK3CA, KDR, and JAK3 genes. In the unpaired CRC (Case 3) that was sequenced, variants in the TP53, CDKN2A, KDR, and JAK3 genes were observed. Specifically, the same variant in the TP53 gene resulting in a codon change (cCc/cGc) and amino acid alteration (P72R) was detected in this CRC, similar to the other CRCs and their corresponding primary breast tumors. In summary, Mahajan and colleagues demonstrated that the breast cancer CRCs analyzed in their study preserved the overall gene mutations, copy number, and miRNA expression patterns of the respective tumor tissue they were derived from [75]. This study highlights the potential of CRCs as a valuable tool for studying various subtypes of breast cancer. 

Ductal carcinoma in situ (DCIS) is a non-invasive form of breast cancer, and currently, there are no reliable predictors to determine its progression into an invasive disease. Therefore, the majority of patients undergo radiation and/or hormone therapy following surgical removal, often resulting in overtreatment. Two commercially available cell lines for studying DCIS, namely SUM225CWN and MCF10DCIS.COM, are not ideal due to their origins that are not derived directly from primary DCIS tumors [99,100,101,102,103]. Five cell lines were generated from a patient in Singapore. These cell lines underwent genetic manipulation through hTERT transfection, a process that enhances their lifespan, but they are not commercially available [103].

Brown et al. sought to develop new models of DCIS by culturing primary DCIS tissue from patients following lumpectomy or mastectomy [74]. Following mechanical and enzymatic dissociation, primary DCIS cells were cultured from 19 patients using CR technology. The resulting cultures consisted predominantly of cytokeratin 8- and EpCAM-positive luminal, as well as cytokeratin 5-, cytokeratin 14-, and p63-positive basal mammary epithelial cells. This composition suggests the maintenance of cellular heterogeneity in the in vitro culture system. Moreover, the cellular identities of these cells were preserved both during the ‘conditionally reprogrammed’ proliferative state and after the withdrawal of conditioned media and the ROCK inhibitor, as indicated by the expression of basal and luminal markers [74]. This study indicates that CR technology can be a valuable and viable model for studying DCIS progression and etiology.

**Table 2 cells-12-02388-t002:** Studies of conditionally reprogrammed cells applied in Breast cancer research.

Origination	Finding	Application	References
Mouse tumor tissue (genetically engineered mouse models of triple negative invasive adenocarcinomas)	CRCs maintain tumor heterogeneity and epithelial cell differentiation.	A model for triple negative mammary cancer	[104]
Human breast tumor tissue	CR breast cancer cells are successfully established and characterized.	in vitro breast cancer mode	[105]
Human breast tumor tissue	CR breast cancer cells at early passages maintain main genetic characteristics of primary tumors.	in vitro breast cancer model	[75]
Human normal mammary tissue	CR enables heterogeneous culture of primary mammary cells.	Establishment of mammary cell line	[97]
Human DCIS tumor tissue	CR DCIS cells are cultured for 2 months expressing both luminal and basal marker and maintaining tumor heterogeneity.	in vitro DCIS model	[74]
Human breast tumor tissue	CR luminal-B breast cancer cells are established in 3 of 5 tissues, demonstrating similar gene expression profile to primary tumors. The CR cells enable the evaluation of drug sensitivity of tamoxifen, docetaxel and adriamycin.	n vitro model of luminal-B breast cancer; drug sensitivity test	[106]
Human Phyllodes tumor of breast tissue	This study demonstrates the feasibility of CR for culturing primary cells for drug discovery, selectively targeting phyllodes tumors of the breast cells.	In vitro model of phyllodes tumors of breast	[107]
Human breast tumor tissue	This study reveals the potential of CRC culture in the detection of CTCs in breast cancer	in vitro breast cancer model	[108]
Human breast tumor tissue	Combining CR and single-cell gene expression analysis enables more precise identification of cancer deregulated genes	in vitro breast cancer model	[109]
Human breast tumor tissue	CR enables detecting high impact-low frequency mutations in primary tumors and metastases	in vitro breast cancer model	[110]
Human tumor and adjacent normal breast tissue	CR enables detecting Heterogeneity in Healthy Normal Breast	in vitro breast cancer and normal mammary model	[111]
Human (male) breast tumor tissue	CR male breast cancer cells are successfully established and characterized.	In vitro model of male breast cancer	[112]

### 4.2. Precision Medicine and Drug Discovery 

Precision medicine is a recently developed strategy for characterizing malignancies biologically and is seen as a new frontier in the treatment and prevention of cancer. Precision medicine has a wide range of applications, including those for prevention, diagnosis, prognosis, monitoring treatment response, and early treatment resistance discovery. Precision medicine aims to do away with the “one size fits all” approach to managing cancer patients. Genetic analysis has recently revolutionized how different malignancies are classified and treated. Precision medicine is built on targeted therapy, a therapeutic approach that uses drugs to specifically target proteins and genes linked to the growth and survival of tumor cells [77]. Due to the lack of suitable in vitro models for breast cancer, this is a major concern in studying drug resistance and treatment response. A specific limitation in recognizing effective drugs for breast cancer is that the results are only based on studies in long-term cultured cell lines or xenograft models; thus, the consequences of most clinical research are usually unsatisfactory. At present, the use of patient-derived models (PDMs) with the features of maintained genotype, high immortality, throughput screening, and xenotransplantation is urgently needed for drug screening, drug discovery, and targeted therapy. The CR method can be used in primary cell cultures from normal and tumor tissues of different types of tissues to retain the genotypic and phenotypic characteristics and heterogeneity of the primary species. CR cells can also be used in cultures under 3D conditions and establishment xenografts in animals (zebrafish or mouse). 

To investigate heterogeneity and drug sensitivity, Mimoto and colleagues generated CRCs from patients with recurrent HR+/HER2- (hormone receptor-positive/human epidermal receptor 2-negative) breast cancer [106]. In CR cells, the pathological features and mutation status were retained, but the RNA expression was different from the original tumor cells. To investigate heterogeneity and drug sensitivity, Mimoto and colleagues generated CRCs from individuals with recurring HR+/HER2- breast cancer. In CR cells, the pathological features and mutation status were preserved, but the RNA expression was different from the original tumor cells. They also evaluated the response of CR cells obtained from a liver metastasis that was ER+/PgR+/HER2- to a total of 224 drugs. Out of these, 66 drugs demonstrated a decrease in cell viability, which included SERD and CDK4/6 inhibitors. The patient initially received SERD and CDK4/6 inhibitors following mastectomy, and for a duration of 13 months, no recurrence was observed. These findings are consistent with the results obtained from the drug screening performed on the CR cells [106].

The findings of this study highlight the potential application of CR cells for drug sensitivity tests and determining appropriate treatments. Their research represents a key step in innovative clinical tools development to aid in decision-making for patients with metastatic HR+/HER2- breast cancer. Phyllodes tumors of the breast are uncommon tumors that involve both stromal and epithelial components. Surgical removal is the primary recommended treatment for Phyllodes tumors. However, despite complete resection, the malignant form of Phyllodes tumors still carries a high recurrence risk, reaching up to 40%. Furthermore, there is currently no consensus regarding the most effective drugs for treating Phyllodes tumors. Recently, Urbaniak et al. applied CR technology to evaluate the response of phyllodes tumor of the breast to seven drugs namely Salinomycin, Bcl-2/Bcl-xL inhibitor ABT-263, paclitaxel, DOX, colchicine, vincristine, and cisplatin. Of these, ABT-263, Salinomycin, and DOX were found to be highly selective toward phyllodes tumor cells [107]. This study revealed the feasibility of using CR technology for drug discovery, selectively targeting phyllodes tumor cells. Based on the metaplastic breast-carcinoma cell line with EGFR amplification from a patient using the CR method, Chung et al. discovered that the combination of EGFR inhibitor and paclitaxel was a promising strategy for metaplastic breast-carcinoma with EGFR amplification [113].

Customizing assays for tumor molecular phenotyping is essential due to variations in the differentiation status of tumors and normal tissues in different patients. In response to this challenge, Anjanappa and colleagues [109] employed a combination of the CR method and single-cell gene expression analysis. This approach aimed to investigate the tumor heterogeneity at an individual patient level. Their study involved 420 tumor cells and 284 adjacent normal cells, focusing on the expression of 93 genes. These genes encompassed the PAM50-intrinsic subtype classifier and genes associated with stemness. Notably, normal and tumor cells marked by ALDH+/CD49f+/EpCAM+ exhibited different clustering compared to unselected normal and tumor cells. Through PAM50 gene-set analysis, they effectively identified both minor and major tumor cell subgroups within the ALDH+/CD49f+/EpCAM+ population. The major clone resembled the tumor’s clinical characteristics. Additionally, by utilizing a gene set linked to stemness, they detected varying activations of stemness pathways within different clones of the same tumor. This refined profiling technique enabled the differentiation between genes truly deregulated in cancer from genes indicating potential precursors of tumor cells. The assays used in their study offer a heightened ability to pinpoint genes deregulated in cancer with greater accuracy [109]. 

Recent progress in DNA sequencing technology have made it feasible to sequence a considerable number of tumor samples at a reasonable cost. Nonetheless, to enhance its clinical applicability, it is crucial to decrease sequencing errors and identify rare mutations that might exist within a small subset of tumor cells. Tumor complexity and intratumor heterogeneity contribute to the diversity of subclones. Despite advancements in next-generation sequencing, pinpointing low frequency mutations in primary tumors and metastases that contribute to subclonal diversity remains a challenge in precision genomics. To address this, Anjanappa and colleagues [110] employed CR technology for short-term cultivation of epithelial cells derived from primary and metastatic tumors. This approach enabled them to expand minor clones and gather epithelial cell-specific DNA/RNA for quantitative next-generation sequencing analysis. Comparative analysis of DNA from unprocessed breast tumors and tumor cells cultured from the same tumors was carried out using the AmpliSeq Comprehensive Cancer Panel. They revealed previously uncharacterized mutations found exclusively in the cultured tumor cells, with some of these mutations reported in brain metastases but not primary breast tumors. Moreover, whole-genome sequencing highlighted mutations enriched in liver metastases across different cancer types. Notably, Notch pathway mutations and chromosomal inversions were detected in all five liver metastases, regardless of the type of cancer. Mutations and rearrangements in the FHIT gene, involved in purine metabolism, were identified in four out of five liver metastases. Additionally, the same set of four liver metastases shared mutations in 32 genes. Among these were mutations in various HLA-DR family members, impacting the OX40 signaling pathway, potentially influencing the immune response to metastatic cells. Pathway analysis of all mutated genes in liver metastases indicated abnormal signaling of tumor necrosis factor and transforming growth factor in these metastatic cells. CR technology employed in their study indicated an improved capacity for identifying mutations in both primary and metastatic cancer cells [110]. Therefore, CR technology offers a new tool for assessing the toxicity and effectiveness of new drugs and developing personalized treatment strategies for breast cancer.

### 4.3. Noninvasive Diagnosis and Surveillance 

Early diagnosis is crucial for effective breast cancer treatment and plays an important role in reducing mortality rates [114]. While mammography is considered the gold standard for imaging breast cancer, its limitations in detecting tumors in dense breast tissue necessitate the use of supplementary detection methods. One commonly employed complementary method is ultrasound. However, ultrasound has its limitations as it may not always detect microcalcifications and can potentially miss early signs of tumors [115]. To achieve early detection of breast cancer, it is imperative to utilize fast, simple, and cost-effective blood-based biomarkers in conjunction with mammography. These biomarkers serve as valuable tools in identifying breast cancer at its early stages.

Liquid biopsies are non-invasive and safe methods for tumor detection. These minimally invasive sampling techniques utilize circulating biomarkers to analyze tumors and deliver reliable diagnostic information. Liquid biopsies encompass various components, such as exosomes [116], microRNA (miRNA) [117], circulating tumor cells (CTCs) [118], circulating tumor DNA [119], etc. CTCs are cells that detach from the primary tumor and enter the bloodstream. Some of these cells manage to evade the body’s immune system and undergo a process known as epithelial-mesenchymal transition. These highly invasive processes, including the acquisition of tumor DNA information, genomic data, and proteomic information, allow for the dynamic monitoring of tumor activity.

In 1869, Thomas R. Ashworth first identified CTCs in the blood of a cancer patient through a comparison of CTC morphology with different tumor cells [120]. Despite their discovery dating back approximately 150 years, there was limited research focused on CTCs until the mid-1990s. This can be attributed to the fact that CTCs are exceptionally rare in the bloodstream, with only a few CTCs present among billions of erythrocytes and millions of leukocytes [121,122]. Hence, the detection of CTCs is technically challenging. Jeong et al. applied CR technology to detect CTCs in breast cancer patients [108]. Their objective was to assess the efficacy of the CR system in detecting CTCs in breast cancer. CTCs were isolated from the peripheral blood of breast cancer patients and cultured using the protocol described by Liu et al. [80]. Subsequently, total RNA was extracted from the cultured CTCs, and reverse transcription PCR (RT-PCR) was performed to amplify the MAGE A1-6 and hTERT genes. RNA extraction was also directly extracted from blood samples, and only RT-PCR was used to analyze the expression of these two genes. Following CRC culture, CTC growth was observed in seven out of the samples (23.3% detection rate). The detection rates of CTCs using RT-PCR for the MAGE A1-6 and hTERT genes in CTCs grown through the CRC culture method were 10.0% and 26.7%, respectively. The positive expression rates for the MAGE and hTERT genes in CTCs assessed only by RT-PCR were 23.5% and 44.1%, respectively. By combining the positive expression rates from RT-PCR alone and CRC culture for the MAGE A1-6 and hTERT genes, the CTC detection rates increased to 23.3% and 53.3%, respectively. Furthermore, when the positive expression rates of the two genes were combined using either method, the CTC detection rate reached its highest value [108]. Their study demonstrated the potential of CRC culture in detecting CTCs in breast cancer. Additionally, the combination of CRC culture and RT-PCR for the MAGE A1 6 and hTERT genes proved to be beneficial in enhancing the detection rate of CTCs in the blood.

### 4.4. Disparity of Breast Cancer

We recently established a CRC-derived biobank that can be used to explore the genetic diversity and in vitro behavior of high risk non-cancer breast epithelial cells. While age-associated decreased viability impacted our collection, we successfully isolated non-cancer high risk ipsilateral breast epithelial cells from a wide range of ages, breast cancer subtypes, and pathological stages [67]. Interestingly, we found that MYC and ribosome related genes were expressed at significantly higher levels in samples from women identifying as black or African-American. Genetic variations linked to geographically defined ancestry have been hypothesized to contribute to the disparity in breast cancer outcome women who identify as black or African-American experience as compared to women who identify as white or Caucasian [70]. To explore if non-cancer ipsilateral breast cells from women who identify as black as compared to white showed any significant differences in gene expression, DEGs between samples obtained from women who self-identified as black (*n* = 6) versus white (*n* = 15) were identified (padj<0.05). One hundred seventy-seven genes were significantly up-regulated and one hundred forty-four genes significantly down-regulated in non-cancer ipsilateral cells from women identifying as black as compared to white (Figure 3A). GSEA analysis of these DEGs showed significant associations with ribosome biology (Figure 3B) and up-regulation of genes associated with MYC (Figure 3C) with MYC itself also higher expressed (padj = 0.052) (Figure 3D). Significantly, the division was not absolute between women identifying as black versus white as a similar pattern of down-regulated ribosome-related and up-regulated MYC-related genes was seen in *n* = 3 women identifying as white (Figure 3D). The methodology used for isolation of cells, preparation of RNA, processing for RNA-seq and GSEA analysis was the same as presented in a previous publication (67). The results suggest that biological differences, particularly Myc-associated pathways, of mammary epithelial cells may contribute to BC disparity, especially BC initiation in addition to social-economic factors. Since Myc induces a multigenic program that involves changes in intracellular calcium signalling and fatty acid metabolism, our previous collaborative study suggested key roles for fatty acid transporters (CD36), lipases (LPL), and kinases (PDGFRB, CAMKK2, and AMPK), each of which contributes to promoting fatty acid oxidation (FAO) in human mammary epithelial cells transduced with Myc [123]. Thus, it is possible that dysregulation of fatty acid metabolism by both biological and social/economic factors contribute disparities of BC in African American population.

Using CR technology, Nakshatri and colleagues [111] found a distinct subgroup of cells in a majority of African American women that exhibited higher CD44 expression but lacked CD24 or EpCAM. These cells displayed elevated expression levels of genes associated with stemness and epithelial-mesenchymal transition. Thus, CR technology allows the studying of basic biological factors of BC disparity using live cells from different populations. Together with traditional genetic and other -omics based studies, these will transform disparity biology studies using live normal or cancer cells from healthy donors and/or BC patients.

### 4.5. Heterogeneity of Breast Cancer

Breast cancer is a category of diseases with high heterogeneity. The heterogeneity is attributed to differences in the genomic, epigenomic, transcriptomic, and proteomic characteristics of the cancer cells, these factors also contribute biological properties of cancer cells such as proliferation, apoptosis, metastasis, and therapeutic response.

Breast cancer heterogeneity can be observed in tumor tissues among individual patients (different patients) or intertumor heterogeneity (different tumors), or intratumor heterogeneity (within the same tumor tissues). To study micro- or molecular heterogeneity of breast cancer, we established cell-derived clones using CR technology (Figure 4A). Nine cell clones were generated from one needle biopsy of breast cancer tissue. Results from whole genome sequencing and transcriptome analyses showed complexity at genomic and transcriptome levels (Figure 4B), while transcriptome analysis indicated two categories of profiling of gene expression in nine single cell-derived clones.

The heterogeneity found within the normal breast can impact the characterization of cancer stem cells. Nakshatri and colleagues [111] employed CR technology to document heterogeneity in healthy breast tissue profiles. Their study involved growing cells from over 60 primary samples using CR system and a combination of nine markers. These markers allowed the quantification of at least 20 cell types per individual. Moreover, the CR method enabled the growth of stem, progenitor, and mature cells, and the percentage of these cell types varied among individuals. They noticed a distinct subgroup of cells in a majority of African American women that exhibited higher CD44 expression but lacked CD24 or EpCAM. This difference in cell population between African American and Caucasian patients was significant. These cells displayed elevated expression levels of genes associated with stemness and epithelial-mesenchymal transition. Notably, this gene expression pattern closely resembled that of PROCR+/EpCAM- mammary stem cells [127]

This result suggests that CR technology may serve as a biological method to study heterogeneity of breast cancer from patients’ specimens, especially minimized samples from advanced breast cancer patients.

## 5. Challenges and Prospects for the Clinical Setting

The CR system has shown great promise in breast cancer research, but it is important to acknowledge and address its limitations. One such limitation is that the CR method lacks the inclusion of essential stromal components, including matrix elements, vascular immune cells, and endothelial cells. This limitation hinders the comprehensive analysis of how stromal cells influence tumor cell growth and how tumor cells respond to drugs [96]. Another challenge is distinguishing tumor cells from normal epithelial cells since sometimes normal cells are generated more than tumor cells such that normal cells may surpass in cultures. However, modifications to the standard CR method can enable selective expansion of tumor cells in vitro [96]. Despite these limitations, CR technology holds excellent application prospects in breast cancer research. There is no perfect model for biomedical research, and researchers must choose an appropriate model that suits their specific research question. In many cases, scientists use a combination of technologies at different levels, from molecules to cells, organs, and populations, for their research goals. We also would like to highlight several aspects for further development in clinical laboratory settings.

## 6. Conclusion

The development of the CR method offers encouraging opportunities for studying breast cancer (Figure 5). Cell cultures from both normal and malignant tissues can be generated quickly and efficiently by using the CR system. These CR cells are particularly noteworthy since they maintain the developmental characteristics of the parental tissue and can regain the ability for cellular differentiation even when conditions are removed. Moreover, CR technology can rapidly produce cultures from small biopsy samples and even cryopreserved tissues as well as from xenografts and organoid tissues. This allows for the creation of cell-derived xenograft tumors and the cultivation of spheroids and organoids, making CR technology a potentially optimal in vitro model for breast cancer research that can facilitate the progress of precision medicine and drug discovery. Moreover, in the future CR technology could help develop precision medicine and could create a living biobank for breast cancer. In short, the use of CR technology provides exceptional opportunities for advancing the prevention, diagnosis, and treatment of breast cancer.

## Figures and Tables

**Figure 1 cells-12-02388-f001:**
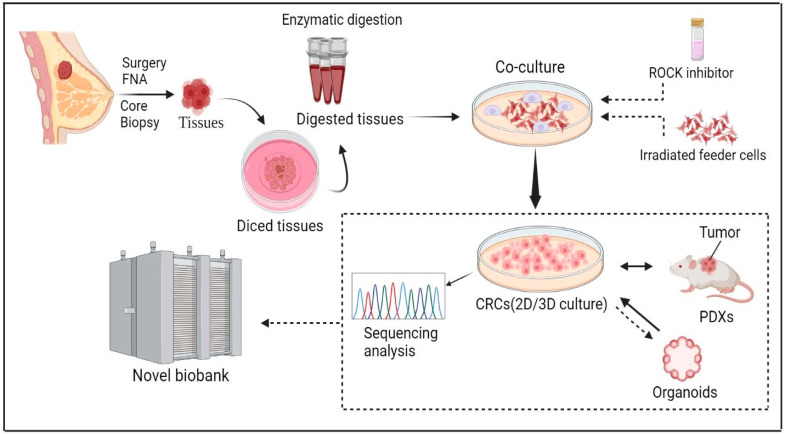
**CR technology in Breast Cancer.** The CR method can quickly generate cultures from normal and cancerous tissue obtained through fine-needle aspiration (FNA), core biopsy, and surgery. Therefore, CR technology can be used as an ideal in vitro model for breast cancer research especially in precision medicine. The figure was drawn using BioRender.

**Figure 2 cells-12-02388-f002:**
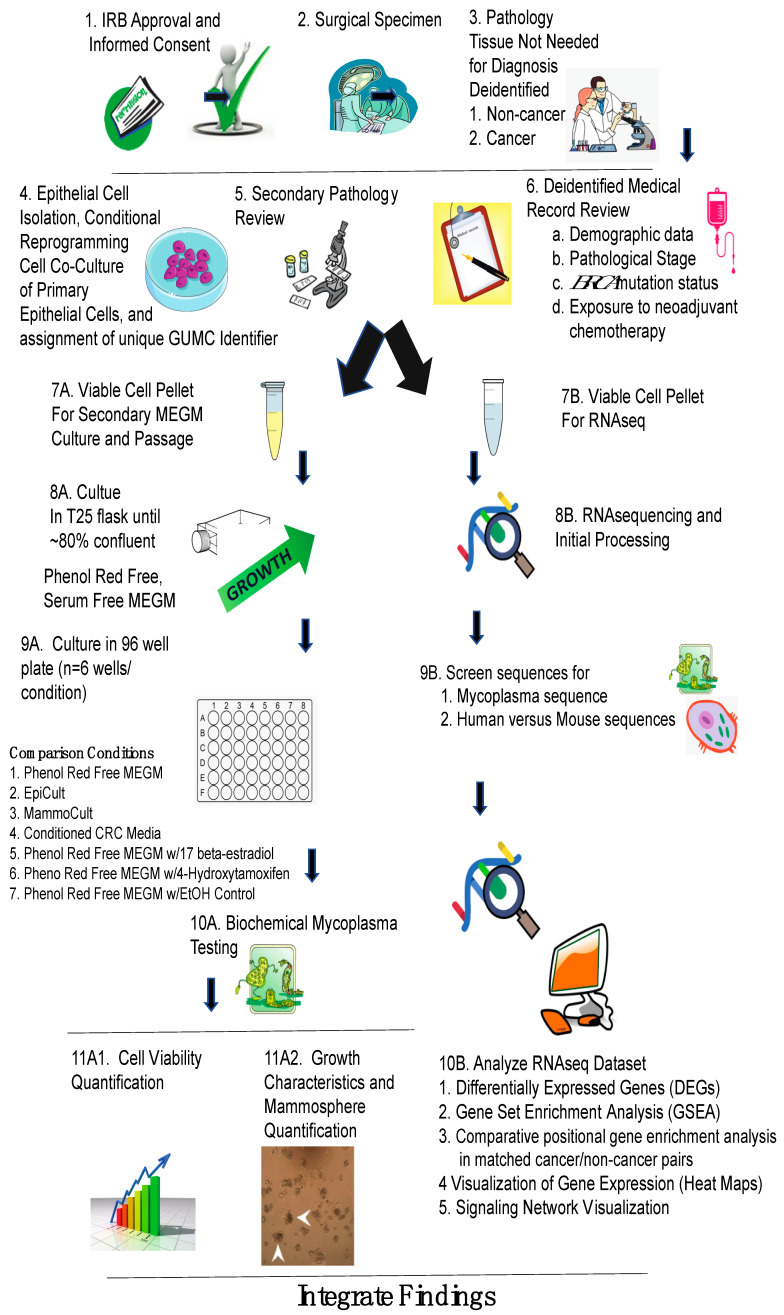
Using a parallel approach to characterize in vitro behavior and transcriptome profiling of CR derived cells. Experimental flow beginning with IRB approval and informed consent (**1**), surgical specimen (**2**), and deidentified tissue acquisition following pathological review (**3**), with initial isolation of mammary epithelial cells using CRC technology with assignment of a unique GUMC identifier to viable cell pellets (**4**), sample validation with secondary pathology review (**5**), and acquisition of deidentified medical records (**6**). At this point sample processing diverged with one viable cell pellet (**7A**) undergoing passage in MEGM (**8A**), followed by culture in a 96 well plate for growth assessment in different media and under different hormonal conditions (**9A**), biochemical mycoplasma testing (**10A**), with analyses of viability (**11A1**), growth characteristics and mammosphere number (**11A2**). A second viable pellet (**7B**) was processed for RNA sequencing (**8B**) with sequence validation for human origin and mycoplasma screening (**9B**) followed by identification of DEGs, GSEA, and visualization of gene expression and signaling networks (**10B**). Abbreviations: IRB: Institutional Review Board. GUMC: Georgetown University Medical Center. MEGM: Mammary Epithelial Cell Growth Medium™. *BRCA*: BReast CAncer gene. RNAseq: RNA sequencing. EpiCult: EpiCult™ Mammary Cell Culture Media. MammoCult: MammoCult™ Human Medium. CRC: Conditionally Reprogrammed Cells. EtOH: Ethanol. n: number. All images Open Source.

**Figure 3 cells-12-02388-f003:**
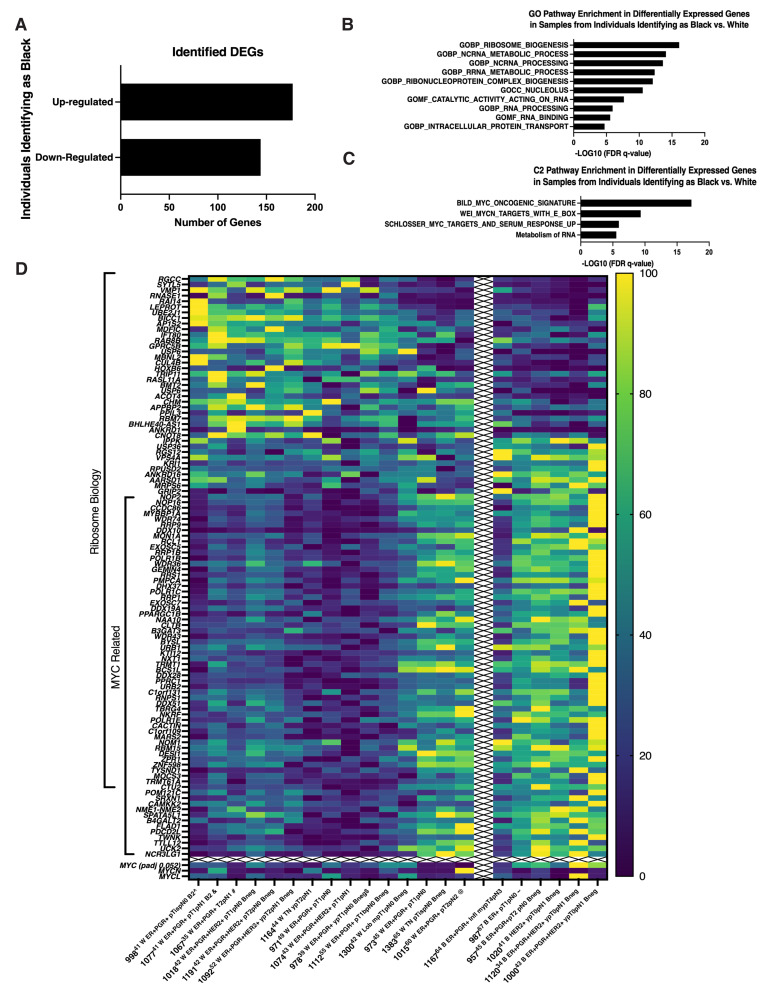
Self-identification as Black or African-American was associated with enrichment in ribosome and MYC biology-related genes. **A**. Bar graph indicating numbers of up-regulated and down-regulated DEGs identified in non-cancer ipsilateral samples from women identifying as Black or African-American versus white. **B.** Bar graph presenting the top ten GO gene sets with the lowest significant FDR q-values identified from the MSigDB Collection utilizing identified DEGs in non-cancer ipsilateral samples from women identifying as Black or African-American versus white. **C.** Bar graph presenting three C2 gene sets with FDR q-values <0.05 identified from the MSigDB Collection utilizing identified DEGs in non-cancer ipsilateral samples from women identifying as Black or African-American versus white. **D.** Heat map illustrating relative expression levels of identified DEGs enriched in samples from individuals identifying as Black or African-American or white. DEGs enriched in Ribosome biology or MYC related are indicated. DEG: differentially expressed gene, at Padj ≤ 0.05. FDR: false discovery rate. MsigDB, Molecular Signatures Database, v7.5.1 [124,125,126] Color coding: Dark blue to yellow with increasing expression.

**Figure 4 cells-12-02388-f004:**
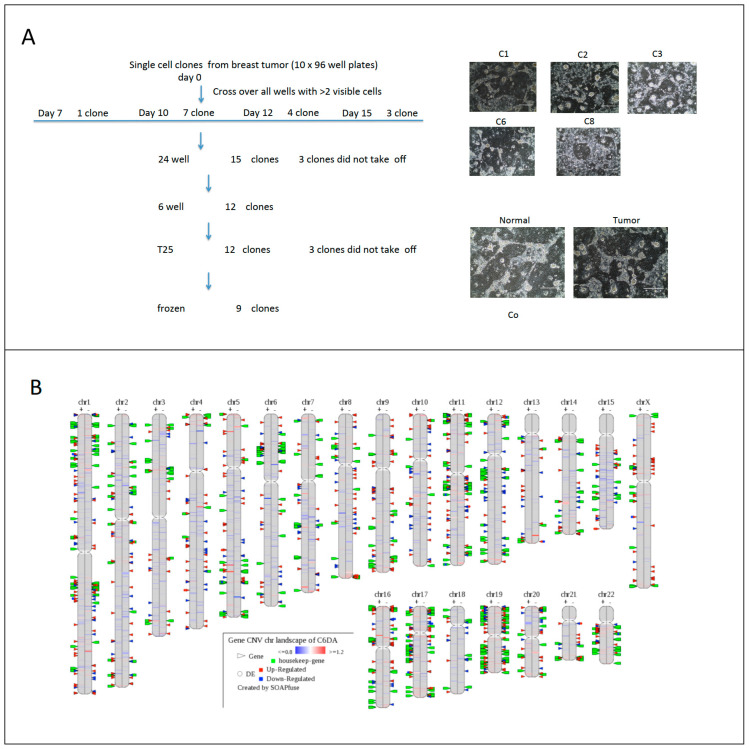
Micro-heterogeneity of breast cancer using CR and single cell derived clones. Needle biopsy from breast cancer tissue was processed and digested as we described previously. Single cell suspension was diluted to 1 cell per 200 uL and 100 uL were plated in each well in ten 96-well plates. All wells were evaluated under light microcopy; wells with two or more than two cells were labeled and left out for the next step. Nine clones were expanded after replating to 24, 12 well-plates and T25 flasks (**A**). DNA and RNA were isolated from 5 clones, adjacent normal and tumor CRCs for whole genome sequencing and transcriptome analyses. CNVs were labeled on the left side of each chromosome of clone number 6 and differential gene expressions were labeled on the right side of chromosome compared to normal cells from same patient (**B**). Correlation of profiling of gene expression of 5 clones indicated two categories of breast cancer cell types (**C**).

**Figure 5 cells-12-02388-f005:**
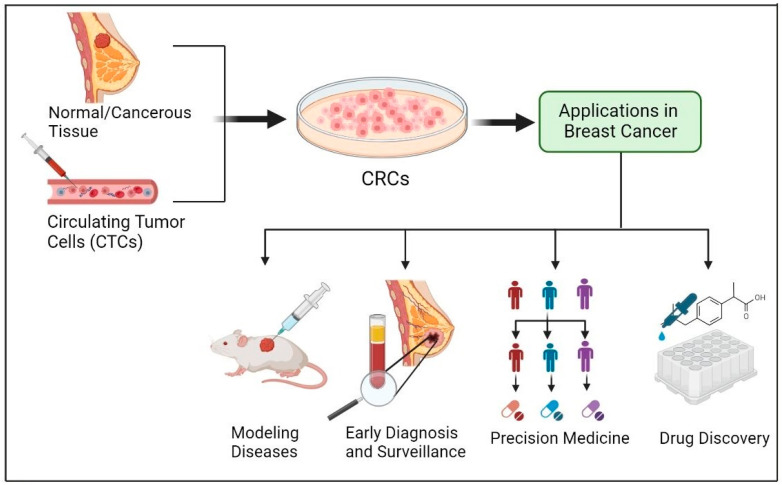
Applications of CR technology in Breast Cancer. CR system enables rapid culture generation from both fresh and cryopreserved normal and malignant tissue samples acquired through surgical procedures, fine-needle aspiration (FNA), and core biopsy. Consequently, CR technology serves as an excellent in vitro model for breast cancer, offering significant potential in drug discovery and precision medicine applications. The figure was drawn using BioRender.

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
