# Peer review of "Unlocking Translational Potential: Conditionally Reprogrammed Cells in Advancing Breast Cancer Research"

_cells, 2023, doi:10.3390/cells12192388_

Round 1

Reviewer 1 Report

The article “Unlocking Translational Potential: Conditionally Reprogrammed Cells in Advancing Breast Cancer Research” is a comprehensive review of the applications of conditional reprogramming technology to preclinical and translational research in breast cancer. The article gives a short explanation of the conditional reprogramming technology, compares more traditional preclinical models of breast cancer with those generated through this technology, and illustrates the various works that have used this technology to study normal breast tissue and its pre-tumoral evolution, as well as breast cancer itself.

A shortcoming of the article is that it seems that the authors have sometimes reported sentences from the abstract or discussion of some papers, without reworking the concepts and making them more understandable and appealing to the reader (e.g., page 8 from line 298, page 17 paragraph from line 585).

There are then some minor issues that should better be changed, such as the following.

Page 1, line 44: the authors refer to breast cancer subtypes as defined according to the variable expression of estrogen and progestin receptors, Ki67 and HER2; it should be acknowledged that this is only a surrogate classification, while the primary classification of breast cancer subtypes is based on gene expression profiling studies.  

Page 2, line 57: primary and acquired resistance do not affect only endocrine therapy but all types of drug treatments for breast cancer.

Page 5, line 204: perhaps it is phenotype instead of phenotypic.

Table 1, the last sentence in the middle column on page 6: please specify to what types of pathogens you refer to.

Page 8, line 272: the word “biology” is repeated twice.

Page 8, line 299: the sentence “a gradual decline in CK19…”, taken from the Discussion of the paper cited as reference 97, refers to works unrelated to conditional reprogramming. The authors should clarify this fact and provide a reference article.

Page 11, lines 345 and 348: there are two sentences starting with “Alamri et al” and “Alamri and colleagues”, respectively, that appear partly a repetition and should better be reworded.

Page 17, line 554: the word efficient is not clear in this context, please explain or reword the sentence.

Page 17, line 585: this paragraph is not clear, e.g., what is “the detection rates of CTCs using RT-PCR for the MAGE A1-6 and 591 hTERT genes in CTCs grown through the CRC culture method” (why searching CTCs by RT-PCR from CTCs?)

Page 20, lines 647-648 are part of the explanation of Figure 3 and must be connected to that.

Page 20, line 664: the definition of intertumor heterogeneity reported here is not, as far as I know, the commonest one; please provide a reference, or use a different definition.

Figure 4, page 21:

the sentences in image A are not clearly readable;

in the figure’s description, line 694, “clonal” number 6 should perhaps be “clone” number 6;

line 696: the sentence is not clear, refers to 6 clones but there appear to be 5 clones.

Page 21, line 712: the sentence is not clear to me, is this an issue that the authors wanted to explore further?

Minor editing could make some sentences more clear.

Reviewer 2 Report

This is an outstanding review by Daneshdoust and colleagues focused on conditionally reprogrammed cells in co-culture systems - - to advance breast cancer research.  My only suggestion is to remove the “References” column in Table 2 and place the references at the end of each “Application”. 

Reviewer 3 Report

The review manuscript submitted by Daneshdoust et al is a very complete text defending the use of Conditionally Reprogrammed Cells on different models of breast cancer research. Although the the text is well written and clear, some modifications are required for its completion. The specific comments are displayed in the .pdf file attached to this review. Overall, with this modifications suggested, the manuscript is adequate for publication.
